# Identification of Sarcopenic Obesity in German Nursing Home Residents—The Role of Body Composition and Malnutrition in the BaSAlt Cohort-Study

**DOI:** 10.3390/nu13113791

**Published:** 2021-10-26

**Authors:** Daniel Haigis, Leon Matting, Silas Wagner, Gorden Sudeck, Annika Frahsa, Ansgar Thiel, Gerhard Eschweiler, Andreas Michael Nieß

**Affiliations:** 1Department of Sports Medicine, University Hospital of Tuebingen, 72076 Tübingen, Germany; silas.wagner@student.uni-tuebingen.de (S.W.); andreas.niess@med.uni-tuebingen.de (A.M.N.); 2Interfaculty Research Institute for Sport and Physical Activity, University of Tuebingen, 72074 Tübingen, Germany; leon.matting@student.uni-tuebingen.de (L.M.); gorden.sudeck@uni-tuebingen.de (G.S.); annika.frahsa@uni-tuebingen.de (A.F.); ansgar.thiel@uni-tuebingen.de (A.T.); 3Institute of Sport Science, Faculty of Economics and Social Sciences, Eberhard Karls University of Tuebingen, 72074 Tübingen, Germany; 4Institute of Social and Preventive Medicine, Faculty of Medicine, University of Bern, 3012 Bern, Switzerland; 5Centre for Geriatric Medicine, Faculty of Medicine, University Hospital of Tuebingen, 72076 Tübingen, Germany; gerhard.eschweiler@med.uni-tuebingen.de

**Keywords:** sarcopenia, obesity, body composition, malnutrition, nursing home

## Abstract

Background: Sarcopenic obesity (SO) is a phenotype, which is defined by reduced muscle strength, muscle mass, and obesity. Limited mobility leads to increased sedentary behavior and decreased physical activity. Both sarcopenia and obesity are aggravated by these factors. In combination, SO is an additional challenge for the setting nursing home (NH). Previous studies have shown a low prevalence of residents with SO in comparable settings, such as community-dwelling. We hypothesize that the BaSAlt cohort also has a small proportion of residents with SO. Methods: For the analysis, 66 residents (women: 74.2%) aged ≥ 65 years from NH, were screened for SO based on EWGSOP2 specifications and cut-off values to classify obesity. Results: Severe sarcopenia was quantified in eleven residents (16.7%). The majority of sarcopenic residents were women (*n* = 10) compared to men (*n* = 1). However, no SO could be identified by assessment of body mass index, fat mass in percentage, and fat mass index. Conclusion: As expected, the setting-specific cohort showed a low number of SO. Furthermore, no case of SO was identified in our study. Sarcopenia was associated with an increased fat-free mass in NH residents. Nevertheless, sarcopenia and obesity play important roles in the preservation of residents’ health.

## 1. Introduction

In 2019, the *European Working Group on Sarcopenia in Older People 2* (EWGSOP2) published the new specifications of assessment methods and an algorithm for the quantification of sarcopenia. Sarcopenia is a musculoskeletal disease associated with loss of muscle strength and muscle mass. If there is also physical functional impairment, severe sarcopenia is present [1]. Bauer et al. (2019) differs this syndrome in primary (age-related) and secondary (disease-related) sarcopenia. This is a key point for the choice of intervention, as they depend on existing comorbidities [2]. Sarcopenia is associated with an increased risk of falls, loss of mobility, and independence. In addition, this musculoskeletal disease has a negative impact on the mental health and social participation of those affected [3]. However, sarcopenia can also be classified into differentiated types. One of those types is sarcopenic obesity (SO). The phenotype is characterized by the presence of sarcopenia as well as obesity [4,5]. The negative effects on physical functioning, morbidity, and mortality are aggravated by their interaction [6]. Studies show different ways to diagnose and treat SO, but there is no consensus [1,7,8]. For the classification of obesity, the determination of the Body Mass Index (BMI) according to the guidelines of the World Health Organization is mainly used to classify obesity [9]. Furthermore, the measurement of fat mass (FM) can be applied. For this, dual-energy x-ray absorptiometry measurements are performed in the clinical setting. A more cost-effective and practical method is the bioimpedance analysis (BIA) [10]. The prevalence of SO in the lifelines cohort study shows a confirmed diagnosis for 0.9% of men and 1.4% of women. It was found that the prevalence increases from the age of 50. Over the age of 70, 82.5% (men) and 80.4% (women) were affected by SO. However, the number of comorbidities and gender plays an essential role. Individuals who had more than three comorbidities showed an increased risk for SO by OR 2.71 (95% CI: 1.62–4.54) in men and 1.33 (95% CI: 1.07–1.65) in women. In contrast, increased physical activity has reverse effects on SO [11].

Previous studies from Germany showed low SO prevalence among older residents in the community-dwelling setting. A gender-specific analysis indicated 2.3% for women and 4.1% for men. However, the cut-off values and the EWGSOP algorithm from 2010 were used for quantification. In addition, the cut-off value of FM was used for the definition of obesity [12,13]. A new evaluation of the SO considering the new EWGSOP2 cut-off values and algorithm is necessary. Moreover, a setting-specific analysis should be performed. Nursing homes (NH) represent one of these settings where limited SO data exist. NH are defined in this context as a long-term care facility with a domestic-styled environment that provides 24-h functional support and health care, as well as daily socialization activities for older persons who require assistance with activities of daily living. Moreover, NH residents often have complex health needs and increased vulnerabilities [14]. The number of comorbidities rises with increasing age. This is associated with the fact that older people require more care. Lastly, independent care can no longer be ensured and the move into the NH follows [15]. Statistics from Germany also show that women represent the majority of NH residents with more than 66% [16].

Furthermore, malnutrition is a frequent problem in the NH setting. It is a multi-etiologic syndrome that involves measurable changes in body function, as well as a direct impact on disease outcomes. The prevalence of malnutrition in NH is 53.4% for a risk of malnutrition and 18.8% for malnutrition, respectively [17]. The correlates of malnutrition are cognitive and physical functional impairments, which are among the common diagnoses in the setting [18]. There is also overlap between sarcopenia and malnutrition by NH residents. Immobility plays a critical role in the development of sarcopenia and malnutrition [19].

Moving into a NH also introduces the risk of developing inactivity and sedentary behavior among residents. The study of physical activity and health of adults in Germany demonstrate increasing sedentary behavior among ages over 70. This additionally promotes the risk of obesity [20]. In the context of sarcopenia, the loss of autonomy in daily life, hand strength, and gait speed after twelve months since moving into the NH are important factors. This must be considered for the prevention and treatment of SO in the setting NH [21]. The need for scientific support of SO in the NH setting is a future field of research.

One aim of the BaSAlt study (Verhältnisorientierte Bewegungsförderung und individuelle Bewegungsberatung im Setting Altenwohnheim -ein biopsychosoziales Analyse- und Beratungsprojekt) is to classify sarcopenia in German NH. From September 2020 to July 2021, the BaSAlt project investigated residents in NH in southwestern Germany in an initial survey. For quantification of sarcopenia according to EWGSOP2 specifications, feasible geriatric assessments were used within the BaSAlt study and methods were implemented to identify SO. We hypothesize that the proportion of SO residents will be low, based on previous studies in comparable settings.

## 2. Methods

### 2.1. Recruitment, Inclusion and Exclusion Criteria

A total of eight nursing homes in southwestern Germany have been included in the study since June 2019. One of them withdrew its participation in the project. The baseline survey t1 took place between September 2020 and July 2021 in seven NH. Due to the prevailing COVID-19 pandemic at the time, the assessments could not be collected by the project team, as contact and visitation bans were in place in the participating facilities. For the surveys, staff of the respective NH were trained for the assessments in a two-day workshop by the BaSAlt team. Supervision and support of the assessors during the geriatric assessments was guaranteed at all times, in accordance with the specified hygiene rules within the facility. A remuneration of 2000€ per NH was provided for the training, recruitment, and implementation of the assessments. Within the measurement time, 14 assessors from seven NH were trained. Two NH withdrew their participation in the assessments after training. This was due to time and structural problems within the facility. Ultimately, the data from five NH were included in the study.

The Ethics Committee of the Faculty of Economics and Social Sciences of the Eberhard Karls University of Tübingen approved the project (no. AZ A2.5.4-096_aa). The BaSAlt study was conducted according to the guidelines of the Declaration of Helsinki. For the recruitment of residents, a study information letter was presented to the participants or their authorized representatives. Permission to inspect the medical file was also requested and could be refused without any disadvantages. All participants or their legal representatives provided written informed consent prior to the study.

Inclusion criteria for the study were a degree of care ≤4 (in the German care system degrees of care are assigned 1–5). Voluntary participation in the study was a prerequisite. The only exclusion criterion was a degree of care 5, which is often associated with bedriddenness and severe physical or mental disabilities of the residents.

### 2.2. Instruments for Demographical and Clinical Data

Residents’ demographic data age (years), sex, and the degree of care (1–4) were obtained from the medical files. The morbidity status was evaluated by categorizing the diseases based on the approved file review into (1) past cardiovascular events, (2) arterial hypertension, (3) coronary heart disease, (4) cardiac insufficiency, (5) cardiac pacemaker, (6) post-stroke/cerebral hemorrhage/TIA, (7) chronic lung disease, (8) cancer, (9) diabetes mellitus II, (10) osteoarthritis of lower extremity, and (11) psychological/emotional/nervous disease of resident. Anthropometric data height (in m), weight (in kg), and BMI (in kg/m^2^) were collected. Additional body composition parameters were recorded as part of the bioimpedance analyzer from Akern (impedance vector analyzer BIA 101 BIVA, 50 kHz ± 1% measuring frequency). Processing of the generated data was performed for further analysis using BodygramPlus Enterprise software (Version 1.2.2.9, Akern s.r.l., Pontassieve, Italy). The body composition via BIA measurement did not determine directly. The body resistance (Rz) and the reactance (Xc) showed their relation in the phase angle (PhA in °). The individual PhA is defined by the arctangent of [Xc/R × 180/π]. Fat mass (FM), fat-free mass (FFM), and muscle mass (MM) in kg and the percentage (FM%, FFM% and MM%) were measured. The fat mass index (FMI) and the fat free mass index (FFMI) adjusted at height (in kg/m^2^) was calculated. Total body water (TBW in l), extracellular water (ECW in l) and body cell mass (BCM in kg) were measured.

Cognitive functioning was assessed using the Mini-Mental Status Test (MMST). Therefore, a maximum of 30 points could be achieved for the MMST. The questions were divided into five categories (orientation, retentiveness, attention and numeracy, recall, and language). A classification of cognitive functioning should be defined as “no dementia” (30–28 points), “mild cognitive impairment” (27–25 points), “mild dementia” (24–18 points), “moderate dementia” (17–10 points), and “severe dementia” (≤9 points) [22].

The Barthel Index (BI) was applied for the survey of the need for care. In the BI, a maximum of 100 points could be scored in ten individual categories of daily living (eating, sitting up and transferring, washing, using the toilet, bathing/showering, getting out and walking, climbing stairs, dressing and undressing, stool control, and urine control). The classification into “completely independent” (100 points), “partially in need of care” (95–85 points), “in need of care” (80–35 points), and “dependent on care” (≤30 points) was scoring by assigning points [23].

In addition, the nutritional status of the residents was collected using the Mini-Nutrition-Assessment Short-Form (MNA-SF©). The MNA-SF© was used to categorize the nutritional status of the residents. Based on six questions (decrease in food intake, weight loss in the last three months, mobility, acute illness or psychological stress in the last three months, neurophysiological problems, and BMI) the assessment was to be performed. A maximum of 14 points could be obtained. The options for categorization as “normal nutritional status” (12–14 points), “risk for malnutrition” (11–8 points), and “malnutrition” (≤7 points) were given [24].

### 2.3. Instruments and Quantification Methodes for SO

Instruments and the algorithm for the quantification of SO were performed according to EWGSOP2 guidelines. Moreover, cut-off values for quantification of sarcopenia and SO were applied based on EWGSOP2 [1] and are illustrated in Table 1.

For case-finding, the SARC-F questionnaire was designed to identify a possible risk of sarcopenia via a subjective self-assessment. Five questions in different areas (strength ability, assistive support when walking, getting up from a chair, climbing stairs, and falls) were determined with the use of a point score. Here, 0–10 points could be achieved in total. For each category 0–2 points could be awarded. A point score ≥ 4 points in total was interpreted as a predictor of sarcopenia [1,25].

To assess the suspicion of sarcopenia, muscle strength testing followed. For the hand force measurement, the maximum force (HFM in kg) of the residents was recorded using an isometric hand force dynamometer (Hydraulic Hand Force Dynamometer Saehan Model SH5001, Saehan, Changwon-si, Korea). Three measurements each were taken alternating the right and left hand. The best trial out of six was used for the maximum force value [26]. The cut-off values according to EWGSOP2 were <16 kg for women and <27 kg for men [1,27].

Confirmation of sarcopenia was checked by muscle mass determination. For this purpose, the measurement of appendicular skeletal muscle mass (ASMM in kg) with BIA was performed [28]. Cut-off values of ASMM were <15 kg for women and <20 kg for men [1,29].

To determine the severity of sarcopenia, the physical functioning of the residents was assessed using the 4-m walking speed test (4MWST in m/s) according to the Short Physical Performance Battery (SPPB) specifications [30]. Habitual gait speed (in m/s) was recorded over a walking distance of four meters and measured using a stopwatch. Before and after the measured distance, run-on and run-off distances of two meters each are considered. EWGSOP2 sets the cut-off value for both sexes at the speed of ≤0.8 m/s [1,31,32]. Residents who were not able to walk, e.g., wheelchair users, were classified as functionally impaired.

Three different cut-off values were chosen for the classification of SO. The BMI was calculated by measuring weight (in kg) and height (in m). The cut-off value of BMI (in kg/m^2^) was ≥30 kg/m^2^ for both sexes [5,9]. Furthermore, FM% was determined based on the BIA measurement. The FM cut-off values (in %) were >42% for women and >30% for men [5,33]. A further assessment was performed by using the FMI, while FM was adjusted for the resident’s height (in kg/m^2^). The cut-off values were ≥13 kg/m^2^ for women and ≥9 kg/m^2^ for men, respectively [10,34].

### 2.4. Data Analysis

For statistical analysis, a descriptive data analysis was performed for the baseline data of the cohort at time t1. Data collection was handwritten using a CRF sheet, which was provided with the respective identification number of the participant. The handwritten data were transferred to the statistical program SPSS (IBM SPSS version 27.0.1.0). Mean values with standard deviation (Mean ± SD) and median values (Md) were also used to represent the group. The significance level was set at *p* ≤ 0.05 for two-sided testing. The calculation of the rank-correlation-coefficient *r_s_* according to Spearman was used for the evaluation.

## 3. Results

For the survey, out of 167 residents, 69 residents (41.3%) from five NH could be recruited at the measurement time t1. Three residents were excluded from the data analysis. One of those residents died during the measurement phase. Another one left the facility and a third resident transitioned to palliative care. A total of 66 residents (74.2% female) were included in the final data analysis. In Table 2, the basic characteristics of the residents are listed.

The degree of care was 3 (Md) for both sexes. Walking aids were used by 69.4% of women and 58.8% of men. In total, 14.3% of women and 23.5% of the men were not able to walk independently. They could only remain mobile in the NH with the support of nursing staff or a wheelchair. In the MMST, 18.4% of women and 17.6% of men could not be categorized. Reasons were blindness (*n* = 6), lack of motivation (*n* = 4), or severe cognitive impairment (*n* = 2). The values of different body composition parameters are shown in Table 3.

The Md of comorbidities among women and men was 3 and 4, respectively. The percentage of women who had more than 3 comorbidities was 38.8% and 64.7% in men. For one resident, access into the resident’s medical file was denied. The categories for morbidity status are reported in Table 4.

EWGSOP2 cut-off values were applied for quantification of sarcopenia. For further identification of obesity, the cut-off values for BMI, FM%, and FMI were determined. Table 5 presents the results of the analysis.

The SARC-F showed difficulties in detecting residents’ risk of sarcopenia. For eight women (16.3%) and two men (11.8%), no assessment was possible due to motivational reasons (*n* = 2), or cognitive limitations (*n* = 8). When HFM was determined, one female and one male resident were unable to complete the test. For both, the assessors’ instructions were not feasible due to their severe cognitive dysfunction.

The analysis showed that eleven of the 66 residents (16.7%) had severe sarcopenia. Women (*n* = 10) were more affected by sarcopenia than men (*n* = 1) at 20.4% and 5.9%, respectively. None of the diagnosed sarcopenic residents showed a SO phenotype. In categories BMI, FM%, and FMI, SO was not confirmed in any of the cases.

The Spearman rank-correlation-coefficient *r_s_* showed negative correlations for sarcopenia between BMI (*r_s_* = −0.37; *p* = 0.002), PhA (*r_s_* = −0.31; *p* = 0.01), FFM (*r_s_* = −0.51; *p* < 0.001), FFMI (*r_s_* = −0.43; *p* < 0.001), MM (*r_s_* = −0.48; *p* < 0.001), TBW (*r_s_* = −0.51; *p* < 0.001), ECW (*r_s_* = −0.38; *p* = 0.002), and BCM (*r_s_* = −0.50; *p* < 0.001). There is no significant correlation in FM, FM%, and FMI (all *p* > 0.05). Moreover, there was no significant correlation between sarcopenia and nutritional status (*p* > 0.05).

## 4. Discussion

The SO is a phenotype of sarcopenia, which is the combination of sarcopenia and obesity. Previous studies show an inconsistent approach to identification by parameters to determine SO. Sarcopenia and obesity lead to an increase in physical functional impairment, morbidities, and an increased risk of mortality. In the case of a combination of sarcopenia and obesity, these negative effects are amplified. Based on the EWGSOP2 standardized algorithm for quantifying sarcopenia in Caucasian populations, this cohort-study identified residents with the musculoskeletal disease in Germany. For this purpose, applicable assessments were performed, which were tested for feasibility in a pilot study [35]. The three sex-specific cut-off values BMI, FM%, and FMI were used to identify the phenotype SO. BIA is a feasible tool to determine the body composition parameters. It is suitable for residents in NH due to its quick and uncomplicated implementation. Our hypothesis assumed a low number of SO residents due to comparable settings in previous studies with similar characteristics.

Severe sarcopenia was found in 16.7% of the residents. The majority were women (90.9%). The prevalence of sarcopenia in long-term care facilities is at 33% [36]. However, these results show an increased prevalence compared to our study, which seems to be due to the change in algorithm and cut-off values in 2019. Some studies have already examined the variation in sarcopenia prevalence with respect to changes in EWGSOP (2010) to EWGSOP2 (2019) [1,31]. A decreasing prevalence of sarcopenia was found [37,38]. Mainly men show deviations in the muscle strength category between EWGSOP and EWGSOP2, with the adjustment of the cut-off value for hand grip strength having a decisive influence on the prevalence (31.9% to 12.0%) [39]. We hypothesize that the changes in cut-off values lead to an underestimation of sarcopenia in the NH setting.

However, no SO could be identified by any of the three cut-off values of obesity. The correlations show a significant association of sarcopenia by reduced BMI, PhA, FFM, FFMI, MM, TBW, ECW, and BCM (*p* < 0.05). Thus, increased FM, FM%, or FMI are not associated with existing sarcopenia in our study (*p* > 0.05). In contrast, a different result can be observed if the BMI is taken into consideration. The BaSAlt cohort demonstrated overweight and obesity in women at 34.7% and 14.3%, respectively. Among men, the proportion of overweight residents is 17.6%. Compared to women, the percentage of obese residents is higher at 29.4%. BMI is not suitable for quantifying SO. This has already been shown by several studies [5,7,8,34]. A feasible method is the BIA vector analysis. Body composition parameters are measured, which are suitable as cut-off values for the determination of the SO. PhA is considered an indicator of membrane integrity and water distribution between the intracellular and extracellular space and thus indirect estimation of body cell mass. The phase angle is becoming increasingly important in the quantification of sarcopenia. The EWGSOP2 also refer in their guidelines to the usefulness of BIA vectors analysis and the PhA as an indicator of sarcopenia [1]. Individuals with decreased PhA show significant associations and increased prevalence for sarcopenia [40]. Moreover, a relationship between PhA and nutrition status is suspected. However, previous studies have not found a reduced PhA to be associated with an increased risk of malnutrition [41]. In the study by Brunani and colleagues (2021), obesity is described by an excess of FM and an increase in ECW. With ageing, a decrease in FFM is additionally reported. This demonstrated that the establishment of reference values in specific age groups could be used as a way to improve the treatment of increased FM or reduced FFM [42]. This should also be pursued for the setting-specific group of residents in NH. In comparison, our study indicated that the sarcopenic residents had a significant decreasing in FFM and ECW. The FM showed no significant increase.

Moreover, the MNA-SF© also showed no association of sarcopenia and nutritional status (*p* > 0.05). Nevertheless, more than 30% of women and more than 47% of men were found to be at risk of malnutrition. Only two of the 49 female residents demonstrated existing malnutrition (4.1%). Among the male residents, no case of malnutrition was identified by MNA-SF©. A systematic review by Eglseer et al. (2016) revealed that for people older than 60 years, sarcopenia correlated with a poor nutritional status. The combined consideration of sarcopenia and nutritional status of the elderly is essential. Factors such as decreased BMI and abnormalities in nutritional screening tools, e.g., the MNA-SF©, must be considered [43]. A meta-analysis by Ligthart-Melis and colleagues (2020) revealed the relationship between sarcopenia and nutritional status. There is a higher probability with OR 4.06 (95% CI: 2.43, 6.80) of developing sarcopenia in older hospitalized individuals as a result of a risk or existing malnutrition [44]. However, our cohort shows a relatively low number of malnourished residents. But more than a third of the female residents and the half of the male residents are at risk of malnutrition and should therefore receive intervention.

The pathogenesis of the SO is a complex construct. SO is affected by the following conditions: (a) insulin resistance conditions the adipose tissue in the body, (b) chronic inflammation and lipotoxicity has a negative effect of the bone, (c) mitochondrial dysfunction and oxidative stress negatively influence the skeletal muscle, (d) and factors such as age, inadequate diet and sedentary lifestyle have a direct impact on the SO. Relevant and demonstrated associations of SO and other comorbidities apply to diabetes type II and cognitive impairment [6,7]. These comorbidities are also found in our study population. When considering the pathogenesis of SO in the context of the setting NH, an increased focus should be placed on the factors inadequate diet and sedentary lifestyle.

Indeed, residents in the BaSAlt cohort-study are not increasingly affected by SO. The novel findings in our BaSAlt cohort-study are an increased occurrence of reduced FFM. In contrast, increasing FM could not be identified. To the best of our knowledge, this is the first time that a survey of SO based on EWGSOP2 specifications has been conducted in German NH. In addition, several parameters (BMI, FM%, and FMI) were used for the analysis of the BaSAlt cohort to identify obesity. An overlap of the phenomena sarcopenia and obesity could not be established. The fact that a large proportion of the residents were also at risk of malnutrition indicates the importance of the sarcopenia intervention.

Nevertheless, an existing overweight or obesity in the NH should also be forced by individual intervention strategies. For the prevention and intervention of sarcopenia, as well as obesity, treatments can be equally effective. But a lack of evidence was reviewed with regard to the effect of muscle strength, muscle mass, and physical functioning with nutrition interventions [45]. In addition, there is a great heterogeneity of the results depending on the investigated group and their age [46]. Adding physical activity and sedentary behavior to the treatment of SO, combinative interventions consisting of diet and physical activity or exercise are adopted. It can be assumed that a combined intervention can achieve greater effects. However, there is inconsistent evidence, which does not allow for a generalized conclusion [47]. Mainly interventions to reduce sedentary lifestyles in NH are of importance, as physical activity or exercise interventions are often difficult to implement. To the best of our knowledge, there are no evidence-based trials in this setting. The implementation of physical activity promotion and the reduction of sedentary behavior in NH is one of the goals of the BaSAlt study [48].

Our study also has limitations. A total of 66 residents from five NH were recruited for the study. Recruitment of residents was managed by the trained nursing staff. However, the implementation of the assessments was limited by time and personnel factors. This is demonstrated by the number of residents participating (69 of 167 residents, 41.3%). To be able to obtain relevant data regarding the prevalence of SO in this setting, a higher number of residents is necessary. For this purpose, a new recruitment in all facilities for the measurement time t2 should be started. In addition, the recruited facilities, which were not involved in the measurement time t1, should be considered. Furthermore, the assessments were performed by facility-specific assessors in the NH. A standardized procedure for the assessments was to be ensured by the uniform training in the survey. The BaSAlt team was in steady contact with the assessors during the measurement time t1 in order to deal with any problems that might arise during recruitment or the assessments. Nevertheless, we assume a selective choice of the recruited residents according to the respective assessors in NH. The available data can therefore only determine a trend in the prevalence of SO.

Due to the COVID-19 pandemic in Germany since March 2020, the planned implementation of the geriatric assessment to quantify sarcopenia in NH of the BaSAlt study was severely limited. Restrictive regulations by government and facility administration as well as local outbreaks of the SARS-CoV-2 virus in some of our cooperating NH made planning and implementation particularly difficult. Moreover, the impact of visitation bans, isolation, and social distancing on residents in our facilities is unclear. However, it can be hypothesized that this has led to increased sedentary behavior and increased inactivity. This results in negative influences on the physical and mental functioning of the residents, which finally has a direct impact on sarcopenia and obesity [16,49].

In summary, further research needs to follow. First, a standard definition of SO must be developed. Furthermore, criteria and cut-off values must be standardized and adjusted according to population and setting. Finally, the prevalence of SO should be determined to formulate further necessary steps for the identification, prevention, and treatment of SO [50].

## 5. Conclusions

Examination of the BaSAlt cohort revealed no identified case of SO in NH residents. However, according to EWGSOP2, sarcopenic residents could be quantified. Furthermore, there was no correlation of increased FM and nutritional status with sarcopenia. For the prevention and treatment of sarcopenia, as well as obesity, interventions can be made via individualized nutritional management and physical activity promotion. Nevertheless, standardized definitions, methods, and cut-off values to identify SO are needed to formulate evidence-based recommendations for practical implementation in the NH setting.

## Figures and Tables

**Table 1 nutrients-13-03791-t001:** Sex-specific cut-off values for SO quantification.

Assessment	Cut-Off-Value for Women	Cut-Off-Value for Men
Hand force measurement [1]	<16 kg	<27 kg
Appendicular skeletal muscle mass [1]	<15 kg	<20 kg
4-m walking speed test [1]	≤0.8 m/s	≤0.8 m/s
Body mass index [2]	≥30 kg/m^2^	≥30 kg/m^2^
Fat mass in % [2]	>42%	>30%
Fat mass index [10]	≥13 kg/m^2^	≥9 kg/m^2^

**Table 2 nutrients-13-03791-t002:** Basic characteristics of the BaSAlt cohort.

	Women (*n* = 49)	Men (*n* = 17)
Age	88.9 (SD ± 6.1)	82.8 (SD ± 8.7)
Height	158.6 (SD ± 7.4)	171.4 (SD ± 6.0)
Weight	65.4 (SD ± 12.3)	80.9 (SD ± 16.0)
Body mass index	26.0 (SD ± 4.9)	27.7 (SD ± 6.5)
<18.5 kg/m^2^	4.1% (*n* = 2)	0.0% (*n* = 0)
18.5–24.9 kg/m^2^	46.9% (*n* = 23)	52.9% (*n* = 9)
25.0–29.9 kg/m^2^	34.7% (*n* = 17)	17.6% (*n* = 3)
≥30.0 kg/m^2^	14.3% (*n* = 7)	29.4% (*n* = 5)
Mini-Mental Status Test	19.0 (SD ± 7.8)	20.6 (SD ± 7.6)
No dementia	8.2% (*n* = 4)	23.5% (*n* = 4)
Mild cognitive impairment	20.4% (*n* = 10)	11.8% (*n* = 2)
Mild dementia	18.4% (*n* = 9)	11.8% (*n* = 2)
Moderate dementia	26.5% (*n* = 13)	35.3% (*n* = 6)
Severe dementia	8.2% (*n* = 4)	0.0% (*n* = 0)
Barthel Index	63.9 (SD ± 23.0)	64.7 (SD ± 25.6)
Completely independent	2.0% (*n* = 1)	11.8% (*n* = 2)
Partially in need of care	24.5% (*n* = 12)	17.6% (*n* = 3)
In need of care	61.2% (*n* = 30)	52.9% (*n* = 9)
Dependent on care	12.2% (*n* = 6)	17.6% (*n* = 3)
Mini-Nutrition-Assessment Short-Form ©	11.7 (SD ± 2.3)	11.7 (SD ± 1.5)
Normal nutritional status	65.3% (*n* = 32)	52.9% (*n* = 9)
Risk of malnutrition	30.6% (*n* = 15)	47.1% (*n* = 8)
Malnutrition	4.1% (*n* = 2)	0.0% (*n* = 0)

**Table 3 nutrients-13-03791-t003:** Body composition of the BaSAlt cohort.

	Women (*n* = 49)	Men (*n* = 17)
Phase angle (PhA in °)	4.9 (SD ± 1.9)	5.1 (SD ± 1.6)
Fat mass (FM in kg)	20.4 (SD ± 9.8)	20.3 (SD ± 13.7)
Fat mass percentage (FM in %)	29.8 (SD ± 10.6)	23.3 (SD ± 11.5)
Fat mass index (FMI in kg/m^2^)	8.1 (SD ± 4.0)	7.1 (SD ± 5.4)
Fat free mass (FFM in kg)	45.0 (SD ± 6.4)	60.5 (SD ± 5.8)
Fat free mass percentage (FFM in %)	70.2 (SD ± 10.4)	76.7 (SD ± 11.5)
Fat free mass index (FFMI in kg/m^2^)	17.9 (SD ± 2.3)	20.6 (SD ± 1.8)
Muscle mass (MM in kg)	18.1 (SD ± 5.8)	29.5 (SD ± 2.5)
Muscle mass percentage (MM in %)	28.2 (SD ± 10.1)	37.6 (SD ± 6.9)
Total body water (TBW in L)	33.6 (SD ± 5.6)	46.1 (SD ± 4.3)
Extracellular water (ECW in L)	17.4 (SD ± 2.4)	23.7 (SD ± 3.5)
Body cell mass (BCM in kg)	21.3 (SD ± 6.6)	29.1 (SD ± 6.2)

**Table 4 nutrients-13-03791-t004:** Categorized comorbidities of the BaSAlt cohort.

	Women (*n* = 49)	Men (*n* = 17)
(1) Past cardiovascular event	24.5% (*n* = 12)	47.1% (*n* = 8)
(2) Arterial hypertension	69.4% (*n* = 34)	64.7% (*n* = 11)
(3) Coronary heart disease	14.3% (*n* = 7)	41.2% (*n* = 7)
(4) Cardiac insufficiency	22.4% (*n* = 11)	58.8% (*n* = 10)
(5) Cardiac pacemakers	6.1% (*n* = 3)	17.6% (*n* = 3)
(6) Post-stroke/cerebral hemorrhage/TIA	26.5% (*n* = 13)	29.4% (*n* = 5)
(7) Chronic lung disease	8.2% (*n* = 4)	11.8% (*n* = 2)
(8) Cancer	12.2% (*n* = 6)	29.4% (*n* = 5)
(9) Diabetes mellitus type II	20.4% (*n* = 10)	29.4% (*n* = 5)
(10) Osteoarthritis lower extremity	26.5% (*n* = 13)	23.5% (*n* = 4)
(11) Psychological/emotional/nervous disease	57.1% (*n* = 28)	64.7% (*n* = 11)

**Table 5 nutrients-13-03791-t005:** Quantification of sarcopenia and obesity in the BaSAlt cohort.

	Women (*n* = 49)	Men (*n* = 17)
**Sarcopenia**		
SARC-F	3.2 (SD ± 2.6)	2.6 (SD ± 2.5)
Risk of sarcopenia	34.7% (*n* = 17)	23.5% (*n* = 4)
No risk of sarcopenia	49.0% (*n* = 24)	64.7% (*n* = 11)
Hand force measurement	13.6 (SD ± 5.1)	23.8 (SD ± 6.3)
Quantified as sarcopenic	57.1% (*n* = 28)	70.6% (*n* = 12)
Quantified as non-sarcopenic	40.8% (*n* = 20)	23.5% (*n* = 4)
Appendicular skeletal muscle mass	16.1 (SD ± 3.7)	22.6 (SD ± 2.3)
Quantified as sarcopenic	30.6% (*n* = 15)	11.8% (*n* = 2)
Quantified as non-sarcopenic	69.4% (*n* = 34)	88.2% (*n* = 15)
4-m walking speed test	0.6 (SD ± 0.2)	0.6 (SD ± 0.2)
Quantified as sarcopenic	83.7% (*n* = 41)	88.2% (*n* = 15)
Quantified as non-sarcopenic	16.3% (*n* = 8)	11.8% (*n* = 2)
**Obesity**		
Body mass index		
Quantified as obese	14.3% (*n* = 7)	29.4% (*n* = 5)
Quantified as non-obese	85.7% (*n* = 42)	70.6% (*n* = 12)
Fat mass in %		
Quantified as obese	8.2% (*n* = 4)	29.4% (*n* = 5)
Quantified as non-obese	91.8% (*n* = 45)	70.6% (*n* = 12)
Fat mass index		
Quantified as obese	12.2% (*n* = 6)	29.4% (*n* = 5)
Quantified as non-obese	87.8% (*n* = 43)	70.6% (*n* = 12)

## Data Availability

Analyzed data sets can be requested from the corresponding author (with justification).

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
