# Peer review of "Identification of Sarcopenic Obesity in German Nursing Home Residents—The Role of Body Composition and Malnutrition in the BaSAlt Cohort-Study"

_nutrients, 2021, doi:10.3390/nu13113791_

Round 1
Reviewer 1 Report
The present manuscript reports the findings of a multi-site investigation of the incidence of sarcopenic obesity, as defined by the European Working Group on Sarcopenia in Older People 2 (EWGSOP2) criteria, among 66 residents from five nursing homes in Germany. The authors report the presence of sarcopenia, primarily among older women, and obesity; however, no study participants met the criteria for sarcopenic obesity. The research question is important and the experimental approach and adaptation to pandemic-imposed restrictions are appropriate. The manuscript is written clearly but would benefit from further editing by a native English speaker to refine grammar and sentence flow.
Specific comments:
Abstract - The second sentence is incomplete. Consider revising the first sentence of the Results subsection to read: "Severe sarcopenia was quantified in eleven residents (16.7%)."
Body - Although all abbreviations are defined, consider writing out the terms in full later in the paper to remind readers what terms the abbreviations represent.
Table 3 - The first column lists % for some outcomes but consider adding units of measurement to the other outcomes (there is plenty of space).
Author Response
The present manuscript reports the findings of a multi-site investigation of the incidence of sarcopenic obesity, as defined by the European Working Group on Sarcopenia in Older People 2 (EWGSOP2) criteria, among 66 residents from five nursing homes in Germany. The authors report the presence of sarcopenia, primarily among older women, and obesity; however, no study participants met the criteria for sarcopenic obesity. The research question is important and the experimental approach and adaptation to pandemic-imposed restrictions are appropriate. The manuscript is written clearly but would benefit from further editing by a native English speaker to refine grammar and sentence flow.
The BaSAlt team would like to thank you for peer-reviewing our manuscript.
We will send the manuscript to two native speakers for revision.
Specific comments:
Abstract - The second sentence is incomplete. Consider revising the first sentence of the Results subsection to read: "Severe sarcopenia was quantified in eleven residents (16.7%)."
We will revise the abstract with the subsection results.
As example: “Severe sarcopenia was quantified in eleven residents (16.7%).The majority of sarcopenic residents were women (n = 10) compared to men (n = 1).”
Body - Although all abbreviations are defined, consider writing out the terms in full later in the paper to remind readers what terms the abbreviations represent.
We will keep the abbreviations in the text. For the tables, we will use the formulated terms.
Table 3 - The first column lists % for some outcomes but consider adding units of measurement to the other outcomes (there is plenty of space).
We will adapt the table in the first column and add the units of measurement of the parameters.
Best regards
Daniel Haigis
Reviewer 2 Report
The authors examined the prevalence of Sarcopenic Obesity in nursing home residents in Germany using EWGSOP2 definition. The quality of presentation is very good, and manuscript was well-written.
Sarcopenic Obesity is a potentially important syndrome but the definition is quite difficult. They approached it scientifically sound methods, but could not find the case in nursing home residents.
The sample size is limited in particular in men. They should examine more case to conclude it.
Minor points
1. Bauer et al. is published 2019 not 2018 officially.
2. Sarcopenia is symptom or syndrome not a "disease". (it is not make sense that age-related sarcopenia is disease.) Please rewrite the term.
3. In introduction, please state about malnutrition in NH residents.
4. What is novel findings in this study?
Author Response
The authors examined the prevalence of Sarcopenic Obesity in nursing home residents in Germany using EWGSOP2 definition. The quality of presentation is very good, and manuscript was well-written.
Sarcopenic Obesity is a potentially important syndrome but the definition is quite difficult. They approached it scientifically sound methods, but could not find the case in nursing home residents.
The sample size is limited in particular in men. They should examine more case to conclude it.
The BaSAlt team would like to thank you for peer-reviewing our manuscript.
We will send the manuscript to two native speakers for revision.
Minor points:
- Bauer et al. is published 2019 not 2018 officially.
We will correct the year of publication.
- Sarcopenia is symptom or syndrome not a "disease". (it is not make sense that age-related sarcopenia is disease.) Please rewrite the term.
We will make a change in terminology.
- In introduction, please state about malnutrition in NH residents.
We will include a paragraph on the relevance of malnutrition in the setting nursing home.
As example: “Also, malnutrition is a frequently problem in the NH setting. It is a multi-etiologic syndrome that involves measurable changes in body function, as well as a direct impact on disease outcomes. The prevalence of malnutrition in NH is 53.4% for a risk of malnutrition and 18.8% for malnutrition, respectively [von Arnim & Wirth, 2018]. The correlates of malnutrition are cognitive and physical functional impairments, which are among the common diagnoses in the setting [Bell et al. 2015]. There is also overlap between sarcopenia and malnutrition by NH residents. Immobility plays a critical role in the development of sarcopenia and malnutrition [Faxén-Irving et al. 2021].”
- What is novel findings in this study?
We will add a paragraph with the findings of our study to the discussion.
As example: “The novel findings in our BaSAlt cohort-study are an increased occurrence of reduced FFM. In contrast, increasing FM could not be identified. To the best of our knowledge, this is the first time that a survey of SO based on EWGSOP2 specifications has been conducted in German NH. In addition, several parameters (BMI, FM%, and FMI) were used for the analysis of the BaSAlt cohort to identify obesity. An overlap of the phenomena sarcopenia and obesity could not be established. The fact that a large proportion of the residents were also at risk of malnutrition indicates the importance of the sarcopenia intervention.”
Best regards
Daniel Haigis